# Verbascoside Elicits Its Beneficial Effects by Enhancing Mitochondrial Spare Respiratory Capacity and the Nrf2/HO-1 Mediated Antioxidant System in a Murine Skeletal Muscle Cell Line

**DOI:** 10.3390/ijms242015276

**Published:** 2023-10-17

**Authors:** Francesca Sciandra, Patrizia Bottoni, Marinella De Leo, Alessandra Braca, Andrea Brancaccio, Manuela Bozzi

**Affiliations:** 1Istituto di Scienze e Tecnologie Chimiche “Giulio Natta”—SCITEC Sede di Roma, Largo F. Vito, 00168 Roma, Italy; 2Dipartimento di Scienze Biotecnologiche di Base, Cliniche Intensivologiche e Perioperatorie, Sezione di Biochimica, Università Cattolica del Sacro Cuore di Roma, Largo F. Vito 1, 00168 Roma, Italy; 3Dipartimento di Farmacia, Università di Pisa, Via Bonanno 33, 56126 Pisa, Italy; 4School of Biochemistry, University of Bristol, Bristol BS8 1TD, UK

**Keywords:** mitochondria, spare respiratory capacity, skeletal muscle, verbascoside, acteoside, heme oxygenase-1 (HO-1), peroxisome proliferator-activated receptor gamma coactivator 1-alpha (PGC-1α), nuclear factor erythroid 2-related factor (Nrf2), antioxidant response

## Abstract

Muscle weakness and muscle loss characterize many physio-pathological conditions, including sarcopenia and many forms of muscular dystrophy, which are often also associated with mitochondrial dysfunction. Verbascoside, a phenylethanoid glycoside of plant origin, also named acteoside, has shown strong antioxidant and anti-fatigue activity in different animal models, but the molecular mechanisms underlying these effects are not completely understood. This study aimed to investigate the influence of verbascoside on mitochondrial function and its protective role against H_2_O_2_-induced oxidative damage in murine C2C12 myoblasts and myotubes pre-treated with verbascoside for 24 h and exposed to H_2_O_2_. We examined the effects of verbascoside on cell viability, intracellular reactive oxygen species (ROS) production and mitochondrial function through high-resolution respirometry. Moreover, we verified whether verbascoside was able to stimulate nuclear factor erythroid 2-related factor (Nrf2) activity through Western blotting and confocal fluorescence microscopy, and to modulate the transcription of its target genes, such as heme oxygenase-1 (HO-1) and peroxisome proliferator-activated receptor gamma coactivator 1-alpha (PGC-1α), by Real Time PCR. We found that verbascoside (1) improved mitochondrial function by increasing mitochondrial spare respiratory capacity; (2) mitigated the decrease in cell viability induced by H_2_O_2_ and reduced ROS levels; (3) promoted the phosphorylation of Nrf2 and its nuclear translocation; (4) increased the transcription levels of HO-1 and, in myoblasts but not in myotubes, those of PGC-1α. These findings contribute to explaining verbascoside’s ability to relieve muscular fatigue and could have positive repercussions for the development of therapies aimed at counteracting muscle weakness and mitochondrial dysfunction.

## 1. Introduction

The skeletal muscle is one of the most energy-demanding tissues, since it must sustain continuous cycles of contraction and relaxation. Albeit to different extents, depending on the composition of the different muscle fiber types, all types of muscle acquire most of their ATP molecules from oxidative phosphorylation, occurring in the mitochondria. In this process, electron carriers, embedded within the inner mitochondrial membrane, transfer electrons from FADH_2_ or NADH to O_2_, in order to produce ATP through the action of the ATP synthase complex. During this redox reaction, a small amount of O_2_ may escape from its complete reduction to H_2_O, forming reactive oxygen species (ROS), especially the superoxide anion (O_2_^·-^), which represent by-products of oxidative phosphorylation. Eukaryotic cells have many enzymes capable of scavenging ROS. For example, the enzyme superoxide dismutase removes the superoxide anion, transforming it into the less reactive and milder oxidant H_2_O_2_. This latter is still able to cross bi-layered biological membranes, producing oxidative damage to different cellular components, including DNA, proteins and lipids, also within the mitochondria. During exercise, the increased energy demand leads to increased oxygen consumption and consequently to the increased production of ROS. It has been demonstrated that a “physiological amount” of ROS is required to trigger an adaptive response to exercise—for example, by activating nuclear factor erythroid 2-related factor 2 (Nrf2), a transcription factor that targets many genes involved in the cellular response against oxidative stress and inflammation [1,2,3,4]. Oxidative damage occurs when ROS production overcomes the scavenging capacity of the antioxidant cellular defenses. ROS accumulation produces mitochondrial damage, resulting in the further increased release of ROS during oxidative respiration. This induces a vicious cycle, which is often associated with mitochondrial dysfunction, leading to muscle loss and muscle weakness, observed in many pathophysiological conditions, including aging-related sarcopenia [5], Duchenne muscular dystrophy [6] and numerous types of myopathies [7,8].

Healthy mitochondria modulate their metabolism in order to maintain proper energy levels inside the cells in different circumstances. Spare respiratory capacity (SRC) represents a functional parameter related to the ability of mitochondria to meet the growing energy needs of the cells when they are subjected to extra work or stressful conditions. Furthermore, SRC reflects the flexibility of cellular metabolism during complex and highly energy-demanding physiological processes, such as cell proliferation, differentiation and apoptosis [9]. A strong reduction in SRC was observed in mdx mice, a model for Duchenne muscular dystrophy [10,11], and in aging [12]. Interestingly, restoration of mitochondrial function alleviates the dystrophic symptoms in mdx mice [13,14].

Verbascoside is a phenylethanoid glycoside, largely found in dicotyledonous plants, which displays significant free-radical-scavenging activity in different cell and animal models [15,16,17]. Verbascoside was proven to increase resistance to fatigue in healthy animal models [18,19] and in mice with induced cachexia, by promoting the removal of dysfunctional mitochondria [20]. In light of these observations, we postulated that verbascoside ameliorates the mitochondrial function in C2C12 cells, which represent a well-established model of skeletal muscle. In order to verify this hypothesis, we evaluated the oxygen consumption rates of murine C2C12 cells, pre-treated with verbascoside, under normal and oxidative conditions, through high-resolution respirometry. In the present study, we also investigated (i) the protective role of verbascoside against oxidative stress induced by H_2_O_2_, (ii) a possible correlation between the protective role of verbascoside and Nrf2 activation and (iii) the effects of verbascoside on the transcriptional levels of the antioxidative enzyme heme oxygenase-1 (HO-1) and the peroxisome proliferator-activated receptor gamma coactivator 1-alpha (PGC-1α), which induces mitochondriogenesis. A deeper comprehension of the molecular mechanisms underlying the beneficial effects elicited by verbascoside on muscles could pave the way for the possible pharmacological use of verbascoside in all physio-pathological conditions characterized by muscle weakness and mitochondrial dysfunction.

## 2. Results

### 2.1. Verbascoside Improved Mitochondrial Spare Respiratory Capacity

In order to investigate the effects of verbascoside on the mitochondrial respiratory function of C2C12 myoblasts and myotubes, we first established a safe working concentration of verbascoside on the basis of a 3-(4,5-dimethylthiazol-2-yl)-2,5-diphenyltetrazolium bromide (MTT) assay performed by treating cells with increasing verbascoside concentrations for 24 h. We opted for a 24 h treatment, since it has been shown that the concentration of a 320 μM verbascoside aqueous solution at pH 7 decreases to 62% after 24 h due to degradation [21]. Verbascoside did not alter C2C12 myotubes’ viability at up to 300 μM, whereas a cytotoxic effect at concentration values equal to or greater than 250 μM was observed on C2C12 myoblasts (Figure 1). Therefore, a concentration of 150 μM was used for the subsequent experiments.

To assess verbascoside’s effects on the respiratory capacity of C2C12 myoblasts and myotubes, high-resolution respiratory (HRR) measurements were carried out on treated and untreated cells, simultaneously, upon the sequential addition of pharmacologic inhibitors of oxidative phosphorylation, such as oligomycin (2.5 µM), uncoupler carbonyl cyanide-4-(trifluoromethoxy) phenylhydrazone (FCCP, 0.05 µM), rotenone (0.5 µM) and antimycin A (2.5 µM). HRR measures, displayed as oxygen flux per cell number, revealed no significant changes in basal oxygen consumption rate (OCR) in both C2C12 myoblasts and myotubes, following verbascoside treatment (Figure 2A,B and Table 1). Similarly, oligomycin-sensitive respiration (proton leak), which is induced by inhibiting ATP synthase with oligomycin and corresponds to resting, non-phosphorylating electron transfer, was not influenced by verbascoside, indicating no alterations of the inner mitochondrial membrane’s integrity or associated proton transport (Figure 2A,B and Table 1). Notably, the level of maximal uncoupled respiratory activity (maximal OCR), recorded in the presence of optimal uncoupler (FCCP) concentrations, was positively influenced by verbascoside in both C2C12 myoblasts and myotubes. This parameter, which is a measure of the functionality of the mitochondrial respiratory system and is independent of the cellular energy demand, was significantly increased by about 27% and 13% in myoblasts and myotubes, respectively, compared to untreated cells (Figure 2A,B and Table 1). Lastly, the evaluation of residual OCR, measured upon the addition of rotenone and antimycin A, which are inhibitors of Complex I and Complex III, respectively, revealed that verbascoside did not change the non-mitochondrial oxygen-consuming processes, for both myoblasts and myotubes (Figure 2A,B and Table 1).

The flux control ratios derived from HRR measures are useful indicators of mitochondrial function as they are internally normalized to the overall number of cells and allow the accurate comparison of treated and untreated cells [22]. Among them, the spare respiratory capacity (SRC), which represents a robust functional parameter to evaluate mitochondrial reserve capacity, was significantly increased by about 51% and 13% in verbascoside-treated myoblasts and myotubes, respectively, compared to untreated cells (Figure 2C,D and Table 2). No differences were reported for the coupling efficiency (CE) and respiratory control ratio (RCR) for both C2C12 myoblasts and myotubes (Table 2).

### 2.2. Verbascoside Partially Restored Mitochondrial Function under Oxidative Conditions

The high-resolution respiratory (HRR) measurements carried out on C2C12 myoblasts and myotubes exposed to oxidative stress revealed that H_2_O_2_ treatment strongly reduced the basal and maximal oxygen consumption rates (OCR), whereas proton leak was slightly reduced in myoblasts and unaltered in myotubes (Table 1). Oxidative conditions did not significantly change the residual OCR, which was due to non-mitochondrial respiration (Table 1). Verbascoside did not rescue the basal OCR, but partially restored the maximal OCR, increasing its values by about 77% in myoblasts and 25% in myotubes, and the spare respiratory capacity (SRC), increasing its values by about 34% in myoblasts and 61% in myotubes, confirming a protective role against oxidative damage (Figure 3, Table 1 and Table 2). In myotubes, verbascoside also restored the RCR by about 66%, indicating greater potential for substrate oxidation and ATP turnover (Table 2).

### 2.3. Verbascoside Protected C2C12 Myoblasts and Myotubes from H_2_O_2_-Induced Reactive Oxygen Species

It has been reported that oxidative stress reduces the spare respiratory capacity (SRC), probably by chemically modifying and damaging some of the proteins involved in the electron transport chain [9]. Therefore, to evaluate the ability of verbascoside to counteract oxidative damage, we pre-treated C2C12 myoblasts and myotubes with increasing concentrations of verbascoside for 24 h and thereafter we exposed the cells to 1 mM H_2_O_2_ for 1 h (myoblasts) and 2.5 h (myotubes) and tested their viability through a 3-(4,5-dimethylthiazol-2-yl)-2,5-diphenyltetrazolium bromide (MTT) assay. As shown in Figure 4A,B, 150 μM verbascoside was able to attenuate the cytotoxic effect induced by H_2_O_2_; indeed, in control myoblasts, exposure to H_2_O_2_ reduced cell viability to 69%, whereas 150 μM verbascoside restored it to 88%. In myotubes, the protective effect was statistically significant also at 75 μM verbascoside (Figure 4B). In particular, the exposure of control myotubes to H_2_O_2_ lowered cell viability to 16%, while 75 μM and 150 μM verbascoside restored cell viability to 38% and 44%, respectively. The ability of verbascoside to counteract the production of H_2_O_2_-induced reactive oxygen species (ROS) was confirmed by a 2′,7′-dichlorofluorescin diacetate (DCF-DA) assay carried out on C2C12 myoblasts and myotubes, pre-treated with increasing concentrations of verbascoside for 24 h. After the addition of DCF-DA, cells were exposed to 1 mM H_2_O_2_ and the fluorescence induced by ROS production was recorded every 15 min. Figure 4C,D show that the anti-ROS activity elicited by verbascoside treatment followed a dose-dependent trend and reached statistical significance at concentration values of 150 μM for myoblasts (Figure 4C) and 75 μM for myotubes (Figure 4D). It is noteworthy that 150 μM verbascoside also reduced the formation of endogenous ROS in myoblasts and myotubes (Figure 4C,D).

### 2.4. Verbascoside Reduced Oxidative Damage by Activating the Nrf2/HO-1 Axis

Nuclear factor erythroid 2-related factor (Nrf2) is a transcription factor that modulates the cellular response against oxidative stress, by inducing the expression of various enzymes with antioxidant activity. We evaluated whether verbascoside stimulated the cellular antioxidant system by regulating the expression levels of Nrf2. Western blotting experiments revealed that 150 μM verbascoside was able to increase the Nrf2 protein levels under oxidative conditions, in both C2C12 myoblasts and myotubes (Figure 5A,B,E,F). Under normal conditions, Nrf2 is maintained in the cytosol by interaction with Keap1, which mediates the ubiquitination-dependent proteasomal degradation of Nrf2. Under oxidative stress, some cysteines of Keap1 undergo chemical modifications, inducing a change in its conformation, which weakens the interaction between Nrf2 and Keap1. Nrf2 escapes the inhibition of Keap1 and moves to the nucleus, where it promotes the transcription of its target genes [23]. In order to verify whether the increased amount of Nrf2 protein was due to its reduced proteasomal degradation, we examined the extent of Nrf2 phosphorylation at serine 40. Indeed, it has been shown that *t*-butylhydroquinone, an antioxidant molecule, promotes the PKC-induced phosphorylation of serine 40, which liberates Nrf2 from Keap1 inhibition [24]. Figure 5C,D shows that 150 μM verbascoside increased the proportion of Nrf2 protein carrying phosphorylation at serine 40 (p-Nrf2) under oxidative stress, although this effect reached statistical significance only in myoblasts, corroborating the Nrf2 protein expression profile.

Since the phosphorylation of serine 40 has no effect on Nrf2 translocation into the nucleus or its activity as a transcription factor [24], we evaluated verbascoside/H_2_O_2_-induced Nrf2 nuclear translocation by confocal fluorescence microscopy. The collected confocal images showed a significant change in the localization pattern of Nrf2 among control untreated cells and verbascoside-treated cells. Namely, in control myoblasts and myotubes, Nrf2 displayed a faint and diffuse signal throughout the cytoplasm and the nucleus. After treatment with 150 μM verbascoside for 24 h under non-oxidative conditions or with 0.5 mM H_2_O_2_ for 1 h, Nrf2 was less disperse and predominantly localized at the perinuclear and nuclear regions, as can be observed in Figure 6A for myoblasts and in Figure 6B for myotubes. In myoblasts exposed to oxidative conditions, pre-treatment with 150 μM verbascoside induced the relevant nuclear translocation of the transcription factor within the nucleus, where the Nrf2 signal was mainly localized, clearly indicating its possible activation (Figure 6A). In myotubes pre-treated with 150 μM verbascoside and exposed to oxidative conditions, the Nrf2 signal was increased in all cellular compartments compared to the control, but its distribution remained uniform throughout the cytoplasm and the nucleus (Figure 6B).

To further investigate the role played by verbascoside in the activation of Nrf2, we analyzed the transcription levels of heme oxygenase 1 (HO-1) and peroxisome proliferator-activated receptor gamma coactivator 1-alpha (PGC-1α), which are two known Nrf2 target genes. An increase in the transcriptional levels of HO-1 and PGC-1α was induced by treating C2C12 myoblasts with 150 μM verbascoside for 24 h, under both normal and oxidative conditions (Figure 7A,C). In C2C12 myotubes, similar results were obtained for the HO-1 transcription levels (Figure 7B), in line with the protection that verbascoside offered not only against H_2_O_2_-induced but even against the endogenous ones. Conversely, the PGC-1α transcription levels were unaffected by verbascoside in C2C12 myotubes (Figure 7D). This could be explained considering that the measurement of mRNA levels was performed on C2C12 cells at the fifth day of differentiation, when the transcription of PGC-1α had already reached its maximum level (Appendix A).

## 3. Discussion

There is an ever-growing number of studies revealing that mitochondrial dysfunction represents a hallmark of many muscular pathologies. Altered mitochondrial structure and function have been observed in induced pluripotent stem cell-derived cardiomyocytes from patients affected by Duchenne muscular dystrophy, an X-linked disease caused by the loss of the dystrophin gene [25]. Interestingly, an abnormal mitochondrial ultrastructure can be observed before the onset of muscle fiber damage in mdx mice, which are a commonly used animal model of Duchenne muscular dystrophy [26,27]. Reduced mitochondrial function has been reported in both myoblasts and myotubes from patients affected by a severe form of congenital muscular dystrophy due to laminin α2 deficiency (MDC1A). Notably, the mitochondrial dysfunction is accompanied by a significant reduction in the expression levels of peroxisome proliferator-activated receptor gamma coactivator 1-alpha (PGC-1α), which plays a crucial role in mitochondrial biogenesis [8]. Mitochondrial impairment also characterizes the muscle tissue of patients affected by Huntington’s disease, a neurodegenerative disorder, whose neurological symptoms often include muscle atrophy [28]. Additionally, in this pathological condition, mitochondrial dysfunction, which can be observed at the very early stages of disease progression, is associated with PGC-1α downregulation [29,30]. Moreover, the accumulation of dysfunctional mitochondria has been proven to significantly contribute to the progressive loss of skeletal muscle mass and strength observed during aging [31]. Given the important contribution of mitochondrial dysfunction to the etiology of a broad range of physio-pathological conditions, the preservation of proper mitochondrial function represents a promising strategy with positive repercussions in several therapeutic areas.

The measurement of the oxygen consumption rate (OCR) is very useful in assessing the functional properties of mitochondria. In this study, we employed the high-resolution respiratory (HRR) measurement technique, in order to evaluate the effects of verbascoside on the mitochondrial respiratory parameters of skeletal muscle cells. HRR measurements were performed on intact C2C12 myoblasts and myotubes under physiological substrate supply by using a coupling control protocol [32], which allows one to evaluate respiratory function without the addition of exogenous substrates and ADP. This method has proven to be particularly suitable for preserving all mitochondrial perturbations affecting cellular respiration [22,32,33,34]. The comparison of the OCR of verbascoside-treated C2C12 myoblasts and myotubes with that of untreated cells revealed, for the first time, that verbascoside increases the maximal OCR of cells and their spare respiratory capacity (SRC), a respiratory parameter, which quantifies the mitochondrial ability to meet a sudden demand for additional energy. Indeed, recent evidence shows that the SRC can be viewed as a marker of mitochondrial fitness, in all organs or tissues requiring a large amount of energy, such as the heart, brain and skeletal muscle. Cells with higher SRC capacity seem to have higher adaptability to stress conditions as compared to cells with low SRC levels [9]. Our results show that verbascoside is also able to partially restore the loss of maximal OCR and SRC displayed by cells subjected to H_2_O_2_-induced oxidative stress (see Table 1 and Table 2). This may contribute to explaining the anti-fatigue action of verbascoside found in other studies [18,19,20].

Many factors contribute to the maintenance of the SRC, such as the availability of mitochondrial substrates produced by the metabolism of nutrients. The mitochondrial homeostasis also plays a crucial role in supporting SRC; for instance, PGC-1α-induced mitochondrial biogenesis has been found to be associated with high SRC values [35], whereas aberrations in PINK/PARKIN-dependent mitophagy, a process aimed at the removal of damaged mitochondria, resulted in lower SRC values [36,37,38]. Interestingly, Zhang and colleagues have recently demonstrated that verbascoside exerts its anti-fatigue activity by promoting mitophagy in an animal model of cachexia [20]. Oxidative stress is another key element that influences mitochondrial integrity and may strongly reduce the SRC values [9]. In our study, verbascoside triggered the nuclear translocation of the nuclear factor erythroid 2-related factor (Nrf2) and the upregulation of its target gene, heme oxygenase-1 (HO-1), thus stimulating an important cellular response against oxidative stress. In myoblasts, but not in myotubes, verbascoside also increased the transcriptional levels of PGC-1α. This suggests that, in myoblasts, verbascoside might elicit an antioxidant effect by activating the PGC-1α/Nrf2/HO-1 signaling pathway. It is noteworthy that the interplay between Nrf2 and PGC-1α generates a loop stimulating a cellular response against oxidative stress [39]. A possible explanation for the lack of PGC-1α modulation in myotubes might arise by observing that, on the fifth day of cell differentiation, when we must harvest cells to extract mRNA, the transcription of PGC-1α had already reached its maximum level, making it difficult to observe further increments (see Appendix A). In the future, it would be interesting to verify whether verbascoside is capable of upregulating PGC-1α in all the pathological conditions in which it is downregulated. Indeed, since it plays a crucial role in mitochondrial biogenesis, any genetic or pharmacological increase in PGC-1α has been proven to restore the normal phenotype in injured mdx mice [40,41] or in HD patients [29].

Our findings are in accordance with previous studies showing that verbascoside is able to preserve mitochondrial membrane potential in different neuronal cell models (i.e., in SH-SY5Y [42], PC12 [43] or primary rat cortical cells [44]), exposed to different drug-induced injuries. Similarly to what we observed, it was proposed that the protective role elicited by verbascoside on Aβ-treated PC12 cells was essentially based on the Nrf2 activation and consequent HO-1 induction [45]. Verbascoside was found to activate Nrf2 also in TNFα-treated A549 lung cells [46], in the neuronal tissue of a zebrafish model of Parkinson’s disease [47], in UV-treated HaCaT keratinocytes [48] and in retinal pigment epithelial cells exposed to high glucose [49].

The protective role of verbascoside against oxidative stress presumably predisposes myoblasts to better regenerative performance. Indeed, the excessive production of ROS leads to apoptosis and inhibits muscle satellite cells’ proliferation [50], whereas counteracting the oxidative stress—for example, by the genetic or pharmacological activation of HO-1—improves the viability and proliferation of myoblasts [51,52]. The increase in the SRC induced by verbascoside treatment might have similar beneficial repercussions, since the differentiation of myoblasts into mature myotubes is accompanied by a metabolic shift from glycolysis to oxidative phosphorylation and strictly depends on mitochondrial function and activity [53,54,55,56]. The stimulation of regenerative processes represents an important therapeutical strategy in many pathological conditions. Indeed, regeneration processes are particularly intense in dystrophic muscles, where they compensate for damage induced by genetic defects, like the loss of dystrophin in Duchenne muscular dystrophy or laminin α2 in MDC1A, but also in muscles undergoing continuous cycles of contraction and relaxation, during intense physical activity.

An interesting possible translational outcome of our study would be to test the use of verbascoside in combination with already consolidated drugs to alleviate the symptoms of muscular dystrophy or other neuromuscular disorders, exploiting available animal models (mdx, myd, dy or others) or even small cohorts of patients.

## 4. Materials and Methods

### 4.1. Cell Culture

The mouse C2C12 myoblast cell line was cultured in growth medium consisting of Dulbecco’s Modified Eagle Medium (DMEM-41966, high glucose, pyruvate, phenol red; Gibco, Life Technologies, Grand Island, NY, USA) supplemented with 10% fetal bovine serum (FBS; Gibco, Paisley, UK), 10 mg/mL penicillin (Sigma–Aldrich, St. Louis, MO, USA) and 10 mg/mL streptomycin (Sigma–Aldrich) and incubated at 37 °C in humidified air with 5% CO_2_. At confluence, myoblasts were induced to differentiate with DMEM supplemented with 5% horse serum (HS; Gibco), 10 mg/mL penicillin (Sigma–Aldrich) and 10 mg/mL streptomycin (Sigma–Aldrich). At days 1, 2, 3, 5 and 7 of differentiation, cells were collected and stored at 80 °C until total RNA isolation. For all experiments, cell lines with passage numbers between 9 and 15 were used.

### 4.2. Cell Viability

C2C12 myoblasts and myotubes were treated with verbascoside (Extrasynthesis, Grenoble, France) at different concentration values ranging between 19 and 500 μM. After 24 h of incubation, control cells and cells treated with increasing concentrations of verbascoside were rinsed twice with serum-free DMEM and then further combined with 1 mM H_2_O_2_ in serum-free DMEM for 1 h (myoblasts) or 2.5 h (myotubes) at 37 °C in humidified air with 5% CO_2_. At the end of the experiments, cell viability was evaluated by the 3-(4,5-dimethylthiazol-2-yl)-2,5-diphenyltetrazolium bromide (MTT) test. Cell cultures were incubated with 0.5 mg/mL MTT reagent, which was transformed into formazan crystals after about 4 h of incubation at 37 °C. The intracellular crystals were solubilized with a solution of 0.04 M HCl in isopropanol. The amount of formazan released into the culture supernatant, which was directly proportional to the number of living cells, was measured using an automatic microplate photometer (PackardSpectracount™, Packard BioScience Company, Meriden, CT, USA) at a wavelength of 562 nm [57]. Each experiment was performed in duplicate and repeated three times; the cell cytotoxicity was calculated according to the following equation:% cell viability = (Sample OD/Control OD) × 100

### 4.3. High-Resolution Respirometry

Respiration in intact C2C12 myoblasts and myotubes was monitored with high-resolution respirometry (Oroboros Oxygraph-2k, Innsbruck, Austria) operating at 37 °C with a 2 mL chamber volume [58]. Cellular respiration experiments were carried out in two O2k chambers operated in parallel after calibration of the oxygen sensors at air saturation and instrumental background correction. Calibration with air-saturated medium was performed immediately before the oxygen flux measurement was taken. The data acquisition and analysis were carried out using the software DatLab, version 4.2 (Oroboros Instruments).

On the day of measurement, C2C12 myoblasts and myotubes, pre-treated with 150 μM verbascoside or vehicle for 24 h, were incubated in serum-free DMEM containing 0.5 mM H_2_O_2_ for 1 h at 37 °C in humidified air with 5% CO_2_. Before the experiment, cells were trypsinized, counted, resuspended in their medium to a final concentration of 1 × 10^6^ cells/mL, added to each Oxygraph chamber (chamber A and chamber B) and thereafter investigated using a phosphorylation control protocol [54]. The experiments began with the measurement of the basal oxygen consumption rate (OCR) (i.e., the respiration of drug-treated cells resuspended in the medium without the addiction of substrates), followed for about 10 min, until a steady-state level was obtained (basal OCR). ATP synthase was inhibited by the addition of oligomycin (2 µg/mL) added to each chamber to detect the OCR from proton leak. The maximal respiration capacity (maximal OCR) was obtained by the addition of small volumes of the uncoupler carbonyl cyanide-4-(trifluoromethoxy) phenylhydrazone (FCCP, 0.25 μM FCCP/step) and the instantaneous observation of its effect on cellular respiration in the uncoupled state. Cell respiration was then measured in the presence of 0.5 μM rotenone, which selectively inhibits CI, and then in the presence of 2.5 μM antimycin A, which inhibits CIII, to estimate the residual OCR. In addition to the instrumental background, the mitochondrial respiration was corrected for the oxygen flux due to residual OCR [58]. Oligomycin-sensitive respiration (ATP-linked OCR) was calculated by subtracting the oligomycin-insensitive respiration rate from basal respiration (basal OCR–proton leak OCR). All inhibitors and uncouplers used in the protocol are able to cross the cell membrane and do not require prior cell permeabilization. The derived respiratory parameters were calculated as follows: assuming that state 3 respiration is equivalent to the rate measured after the addition of FCCP (‘state 3_FCCP_’or ‘state 3u’) and state 4 is the rate measured after the addition of oligomycin (‘state 4_oligomycin_’or ‘state 4o’), the uncoupled respiratory control ratio (RCR) was calculated by dividing the state 3u rate by the state 4o rate (maximal OCR/proton leak OCR) [59]. The spare respiratory capacity (SRC) was derived by calculating the ratio between the maximal capacity of the electron transport system and the basal OCR (maximal OCR/basal OCR). The coupling efficiency (CE), which represents the proportion of mitochondrial oxygen consumption used to synthesize ATP, was calculated by dividing the oligomycin-sensitive respiration rate by the basal OCR (ATP-linked OCR/basal OCR) [22]. Each experiment was carried out in duplicate and repeated four times.

### 4.4. Detection of Reactive Oxygen Species

The production of intracellular reactive oxygen species (ROS) was evaluated using a 2′,7′-dichlorofluorescin diacetate (DCF-DA)–cellular ROS detection assay kit (Abcam, Cambridge, UK). Briefly, C2C12 myoblasts and myotubes were treated with increasing concentrations of verbascoside, ranging from 19 μM to 150 μM, for 24 h. According to the manufacturer’s instructions, after the addition of DCF-DA, cells were treated with PBS containing 1 mM H_2_O_2_. DCF-DA is initially a non-fluorescent compound, which is deacetylated by cellular esterases to a non-fluorescent compound and later oxidized by ROS into 2′,7′-dichlorofluorescein (DCF), a highly fluorescent compound. Fluorescence was quantified every 15 min for 1 h, after the addition of 1 mM H_2_O_2_ in PBS, using a multi-well plate reader (Promega, Madison, WI, USA), at excitation/emission 485/535 nm. ROS production was expressed as the fluorescence intensity. Each experiment, performed in duplicate, was repeated three times.

### 4.5. Protein Extraction and Western Blotting Analysis

C2C12 myoblasts and myotubes, pre-treated with 150 μM verbascoside or vehicle for 24 h, were incubated in serum-free DMEM containing 0.5 mM H_2_O_2_ for 1 h at 37 °C in humidified air with 5% CO_2_. Subsequently, cells were scraped in cold PBS buffer, centrifugated at 1000 rpm for 5 min and resuspended in lysis buffer (150 mM NaCl, 1% Triton X-100, 50 mM Tris HCl pH 8.0) containing protease and phosphatase inhibitors (Roche, Basel, Switzerland). Protein concentrations were measured by the Bio-Rad protein assay (Bio-Rad Laboratories, Hercules, CA, USA). Samples were denatured in SDS–PAGE sample buffer for 5 min at 99 °C, loaded onto a 10% SDS PAGE gel (BioRad) and transferred to a PVDF membrane. After transfer, the membrane was blocked for 2 h at room temperature with 3% BSA (Sigma–Aldrich) in TBS containing 0.1% Tween 20 (Sigma–Aldrich). PVDF membranes were probed with rabbit polyclonal anti-Nrf2 (1:1000, #16396-1-AP, Proteintech, Planegg-Martinsried, Germany), rabbit polyclonal anti-phospho-Nrf2 (Ser 40) (1:1000, #PA5-67520, Invitrogen, Waltham, MA, USA) or anti-tubulin-HRP (1:1000, sc-23949, Santa Cruz, CA, USA), as tubulin was used as a loading control protein. Signals were captured with an Alliance Q9-atom manual imaging system (Uvitec, Cambridge, UK), using an enhanced chemiluminescence system. Densitometric analyses were carried out with the software Nine-Alliance, version 9.7 (Uvitec). Each experiment was repeated three times.

### 4.6. RNA Isolation and RT-PCR

Before RNA extraction, C2C12 myoblasts and myotubes were pre-treated with 150 μM verbascoside or vehicle for 24 h and then incubated in serum-free DMEM containing 0.5 mM H_2_O_2_ for 1 h at 37 °C in humidified air with 5% CO_2_. Total RNA was isolated from C2C12 myoblasts and myotubes using the RNeasy Mini Kit (Qiagen, Hilden, Germany) according to the manufacturer’s instructions. An additional on-column DNase treatment was performed to remove residual DNA. Then, 1 μg of total RNA was reverse-transcribed in a 20 μL reaction mixture using the High-Capacity cDNA Reverse Transcription Kit (Applied Biosystems, Foster City, CA, USA), following the manufacturer’s instructions. Quantitative Real-Time PCR was performed using a standard TaqMan^®^ PCR protocol on a StepOne Real-Time PCR System (Applied Biosystems) with probes specific for murine heme oxygenase 1 (HO-1) and peroxisome proliferator-activated receptor gamma coactivator 1-alpha (PGC-1α). The housekeeping gene hypoxanthine phosphoribosyltransferase (HPRT) was used as a reference. The reactants were incubated at 50 °C for 2 min and 95 °C for 10 min, followed by 40 cycles of 95 °C for 15 s and 60 °C for 1 min. All reactions were run in triplicate. The relative level for each gene was calculated using the 2^−ΔΔCT^ method [60] and reported as a fold change percentage. Each experiment was performed in triplicate and repeated three times.

### 4.7. Immunofluorescence

C2C12 myoblasts and myotubes, pre-treated with 150 μM verbascoside or vehicle for 24 h, were incubated in serum-free DMEM containing 0.5 mM H_2_O_2_ for 1 h at 37 °C in humidified air with 5% CO_2_. Subsequently, cells were fixed with 4% paraformaldehyde for 10 min at room temperature. Non-specific sites were blocked with 1% BSA in TBS containing 1% Triton X-100. Cells were incubated with the anti-Nfr2 polyclonal antibody (1:50, #16396-1-AP, Proteintech, Germany) for 1 h. Cells were washed with TBS and then incubated with an anti-rabbit secondary antibody conjugated with rhodamine (1:500, ThermoFisher, Waltham, MA, USA) and imaged with a confocal laser scanning system (A1+, Nikon, Tokyo, Japan) using laser excitation at 562 nm to collect emission signals from rhodamine. DAPI staining was used to visualize nuclei [61]. Analysis of images was carried out using the software ImageJ, version 2.9.0. Fluorescence intensity analysis of 50 cells at the same focus plane was performed along the same distance by the ROI Manager tool.

### 4.8. Statistical Analysis

Data are presented as mean values ± standard deviation (SD). Student’s *t*-test was used throughout this study for statistical analyses, assuming equal variance. *p*-values were calculated based on the 2-tailed test; a *p* value < 0.05 was considered statistically significant. Statistical analysis was performed with the software GraphPad Prism, version 8.0.2.

## 5. Conclusions

In conclusion, we found, for the first time, that verbascoside enhanced the maximal oxygen consumption rate and mitochondrial spare respiratory capacity in murine myoblasts and myotubes. Furthermore, our data suggest that the verbascoside-induced activation of the Nrf2/HO-1 axis might be related to the observed reduction in endogenous and H_2_O_2_-induced ROS levels and increased cell viability under oxidative stress conditions. The cytoprotective activity of verbascoside against oxidative stress could thus contribute to improving mitochondrial function. Our findings suggest a novel mechanism underlying the beneficial effects elicited by verbascoside, which include anti-fatigue action. Further work will be necessary to investigate in more detail the specific molecular mechanism behind verbascoside’s action. The results here presented, in fact, do not rule out the possible presence of alternative metabolic or genetic pathways, yet unidentified, also responsible for the protective effect exerted by verbascoside in skeletal muscle cells.

## Figures and Tables

**Figure 1 ijms-24-15276-f001:**
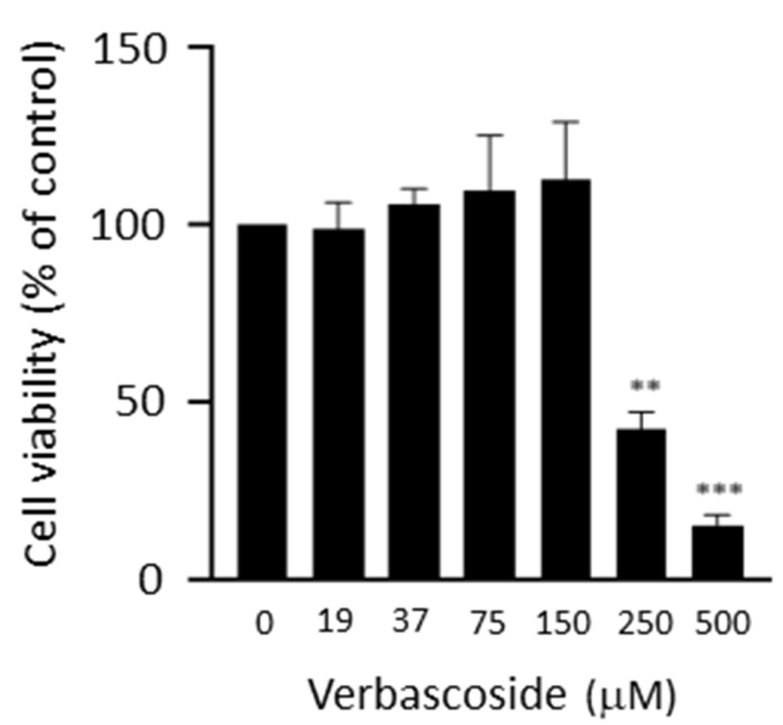
Cell viability was measured using 3-(4,5-dimethylthiazol-2-yl)-2,5-diphenyltetrazolium bromide (MTT) tests. C2C12 myoblasts were treated with increasing concentrations of verbascoside for 24 h. Data are reported as mean percentage values ± SD relative to control (100%) of three independent experiments performed in duplicate. Statistical analysis was performed using Student’s *t*-test (** *p* ≤ 0.01, *** *p* ≤ 0.001, verbascoside-treated cells vs. untreated cells).

**Figure 2 ijms-24-15276-f002:**
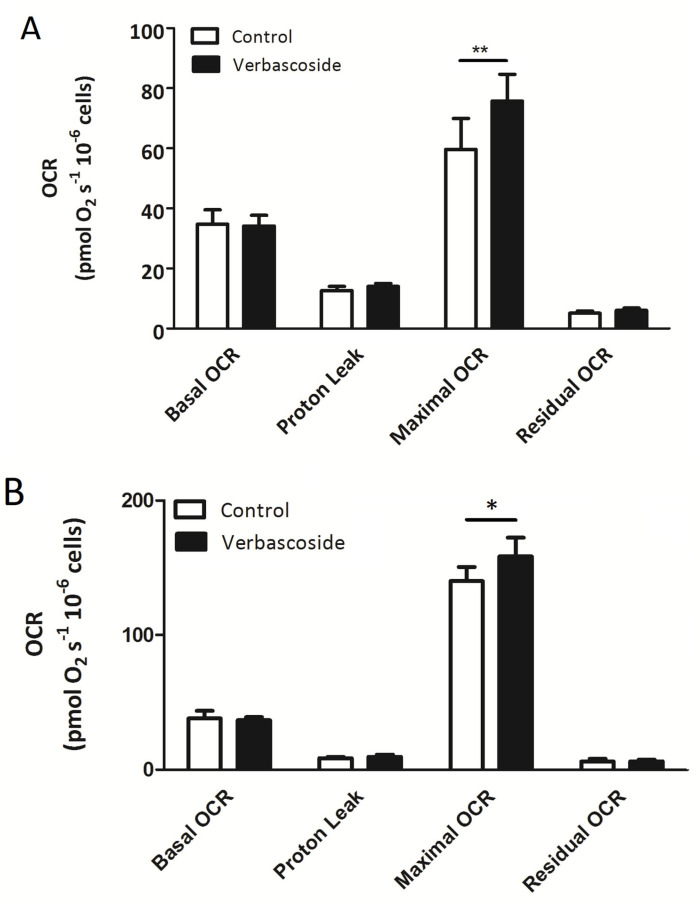
High-resolution respirometry measurements carried out on C2C12 myoblasts (**A**) and myotubes (**B**) treated with 150 μM verbascoside for 24 h. Basal oxygen consumption rate (OCR), proton leak, maximal OCR and non-mitochondrial respiration (residual OCR) are expressed as [pmol/(s × 10^6^)] and are average values ± SD of four independent experiments performed in duplicate. The OCR obtained after addition of 0.5 μM rotenone and 2.5 μM antimycin A (residual OCR) was subtracted from all other OCRs. Statistical analysis was performed using Student’s *t*-test (* *p* ≤ 0.05, ** *p* ≤ 0.01). The spare respiratory capacity (SRC) was calculated as the ratio between maximal OCR and basal OCR and is reported as mean ± SD value of four independent experiments performed in duplicate on C2C12 myoblasts (**C**) and myotubes (**D**). Statistical analysis was performed using Student’s *t*-test (* *p* ≤ 0.05, ** *p* ≤ 0.01).

**Figure 3 ijms-24-15276-f003:**
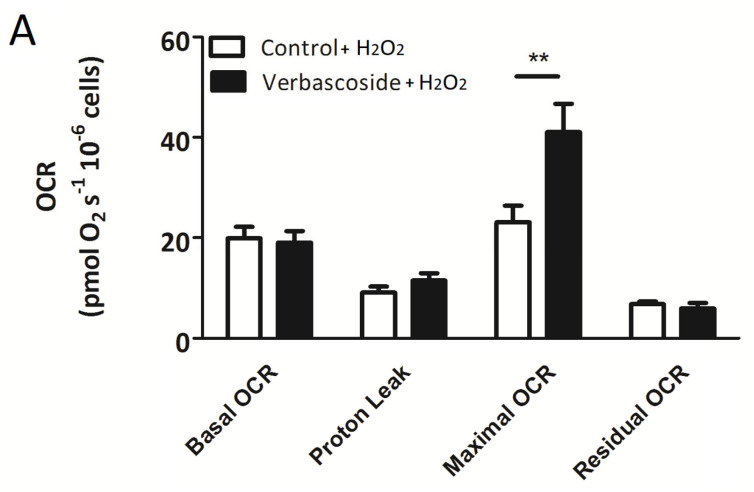
High-resolution respirometry measurements carried out on C2C12 myoblasts (**A**) and myotubes (**B**) pre-treated with 150 μM verbascoside for 24 h and exposed to 0.5 mM H_2_O_2_ for 1 h. Basal oxygen consumption rate (OCR), proton leak, maximal OCR and non-mitochondrial respiration (residual OCR) are expressed as [pmol/(s × 10^6^)] and are average values ± SD of four independent experiments performed in duplicate. The oxygen consumption rate obtained after addition of 0.5 μM rotenone and 2.5 μM antimycin A (residual OCR) was subtracted from all other OCRs. Statistical analysis was performed using Student’s *t*-test (* *p* ≤ 0.05, ** *p* ≤ 0.01). The spare respiratory capacity (SRC) was calculated as the ratio between maximal OCR and basal OCR and is reported as mean ± SD value of four independent experiments performed in duplicate on C2C12 myoblasts (**C**) and myotubes (**D**). Statistical analysis was performed using Student’s *t*-test (* *p* ≤ 0.05).

**Figure 4 ijms-24-15276-f004:**
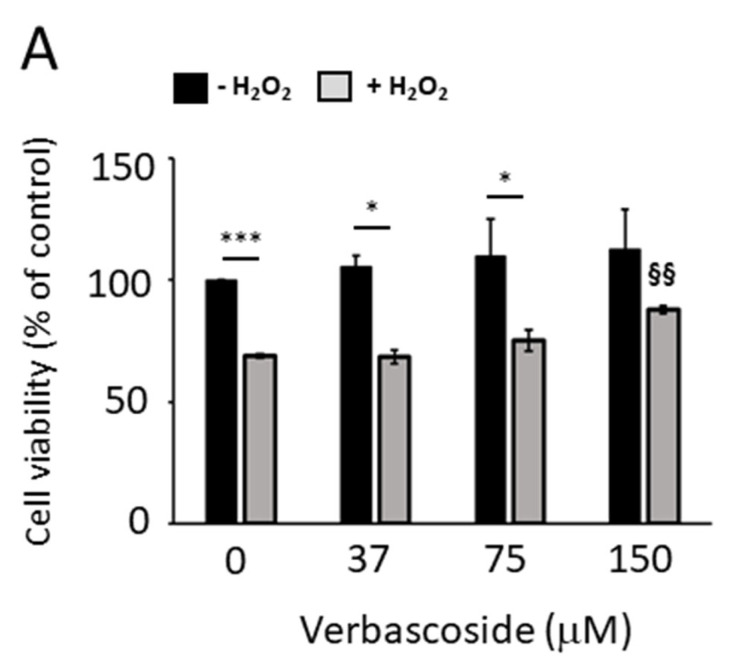
Cell viability was measured using 3-(4,5-dimethylthiazol-2-yl)-2,5-diphenyltetrazolium bromide (MTT) tests. C2C12 myoblasts (**A**) and myotubes (**B**) pre-treated with increasing concentrations of verbascoside for 24 h were exposed to 1 mM H_2_O_2_ for 1h (myoblasts) and 2.5 h (myotubes). Data are reported as mean percentage values ± SD relative to control (100%) of three independent experiments performed in duplicate. Statistical analysis was carried out using Student’s *t*-test (* *p* ≤ 0.05, *** *p* ≤ 0.001 H_2_O_2_-treated cells vs. untreated cells at the same concentration of verbascoside; §§ *p* ≤ 0.01, §§§ *p* ≤ 0.001 verbascoside and H_2_O_2_-treated cells vs. H_2_O_2-_treated cells). Intracellular reactive oxygen species (ROS) induce oxidation of 2′,7′-dichlorofluorescin, resulting in a fluorescent compound whose concentration is directly proportional to the amount of ROS. C2C12 myoblasts (**C**) and myotubes (**D**) were pre-treated with increasing concentrations of verbascoside and exposed to 1 mM H_2_O_2_. Fluorescence intensity is expressed in arbitrary units (A.U.) and reported as average values ± SD from three independent experiments performed in duplicate. Statistical analysis was carried out using Student’s *t*-test (** *p* ≤ 0.01, *** *p* ≤ 0.001, H_2_O_2_-treated cells vs. untreated cells at the same concentration of verbascoside; § *p* ≤ 0.05, §§ *p* ≤ 0.01, verbascoside and H_2_O_2_-treated cells vs. H_2_O_2-_treated cells; **#** *p* ≤ 0.05, verbascoside-treated cells vs. untreated cells).

**Figure 5 ijms-24-15276-f005:**
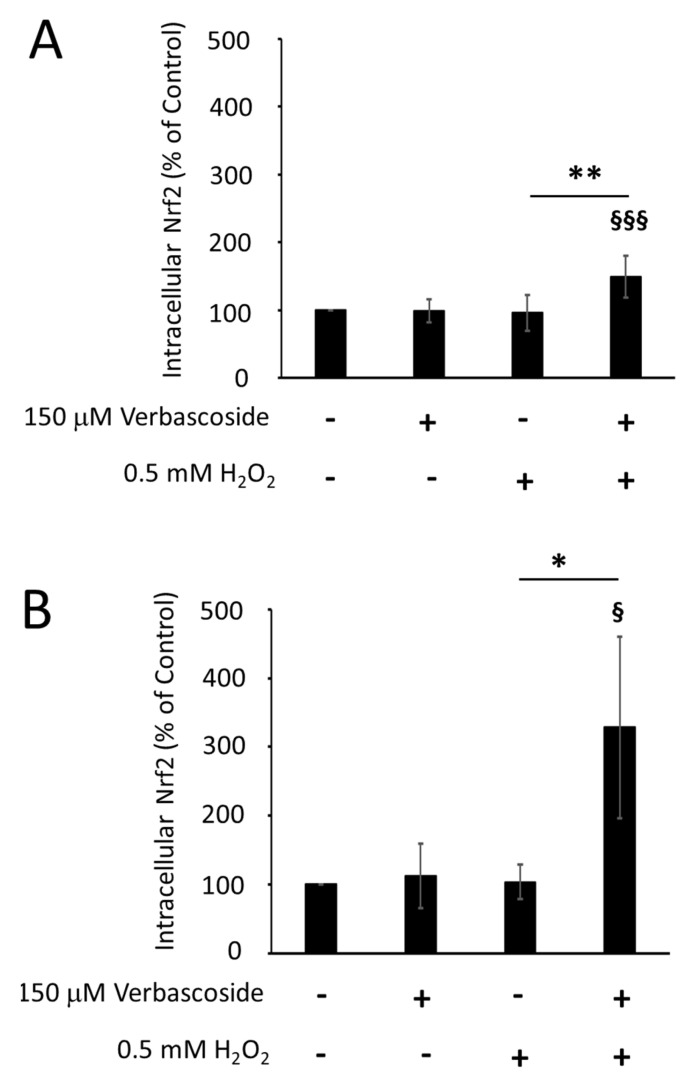
Pre-treatment of cells with 150 μM verbascoside for 24 h, followed by exposure to 0.5 mM H_2_O_2_ for 1 h, increased the expression levels of nuclear factor erythroid 2-related factor (Nrf2) in C2C12 myoblasts (**A**) and myotubes (**B**) and the p-Nrf2/Nrf2 ratio in C2C12 myoblasts (**C**) and myotubes (**D**). Data are expressed as the mean percentage relative to untreated cells (control 100%) ± SD of at least three independent experiments. Differences between mean values were assessed by Student’s *t*-test (* *p* ≤ 0.05, ** *p* ≤ 0.01, verbascoside-treated vs. untreated cells under oxidative conditions; § *p* ≤ 0.05, §§§ *p* ≤ 0.001 verbascoside and H_2_O_2_-treated cells vs. untreated cells). Representative Western blotting experiments showing protein expression levels of Nrf2 and p-Nrf2 in C2C12 myoblasts (**E**) and myotubes (**F**). Tubulin was used as a loading control.

**Figure 6 ijms-24-15276-f006:**
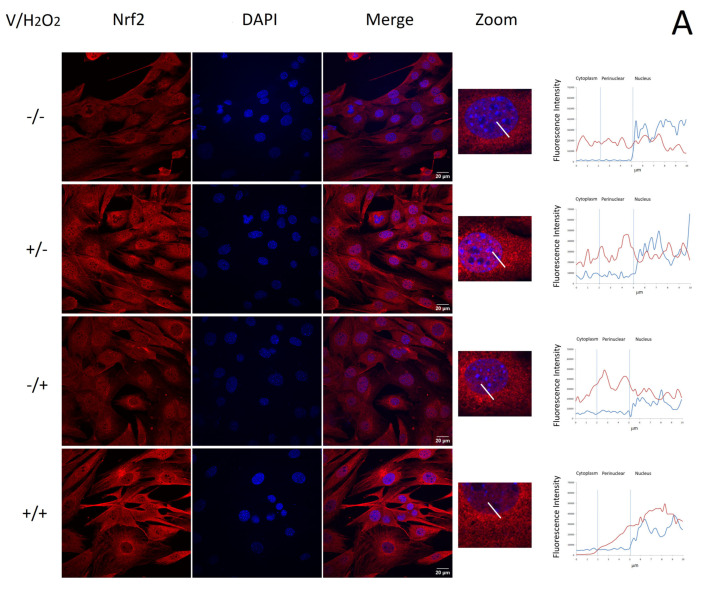
Representative confocal immunofluorescence images of C2C12 myoblasts (**A**) and myotubes (**B**) treated with 150 μM verbascoside for 24 h, exposed to 0.5 mM H_2_O_2_ for 1 h or exposed to 0.5 mM H_2_O_2_ for 1 h after treatment with 150 μM verbascoside for 24 h. Fluorescence intensity scans (panels on the right) along the white bar reported in the merged and zoomed images were acquired using the ROI Manager tool (*n* = 50). Nuclear factor erythroid 2-related factor (Nrf2) signal in red, nucleus in blue (scale bar 20 μm).

**Figure 7 ijms-24-15276-f007:**
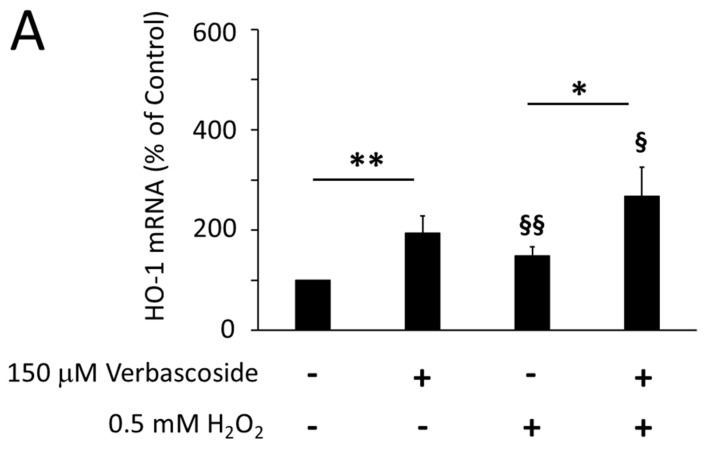
Pre-treatment of cells with 150 μM verbascoside for 24 h increased the transcription levels of heme oxygenase-1 (HO-1) in C2C12 myoblasts (**A**) and myotubes (**B**) and of peroxisome proliferator-activated receptor gamma coactivator 1-alpha (PGC-1α) in C2C12 myoblasts (**C**) but not in myotubes (**D**), under both oxidative and non-oxidative conditions. mRNA levels were evaluated by Real-Time PCR and are expressed as the mean percentage relative to untreated cells (control 100%) ± SD of three independent experiments run in triplicate. Differences between mean values were assessed by Student’s *t*-test (* *p* ≤ 0.05, ** *p* ≤ 0.01, *** *p* ≤ 0.001, verbascoside-treated cells vs. untreated cells; § *p* ≤ 0.05, §§ *p* ≤ 0.01, §§§ *p* ≤ 0.001 verbascoside and H_2_O_2_-treated cells vs. untreated cells).

**Table 1 ijms-24-15276-t001:** Oxygen consumption rates (OCR) are expressed as [pmol/(s × 10^6^)] and are average values ± SD from four independent experiments performed in duplicate on C2C12 myoblasts and myotubes. Statistical analysis was performed using Student’s *t*-test (* *p* ≤ 0.05, ** *p* ≤ 0.01, verbascoside-treated (+V) vs. untreated cells (−V)).

**Myoblasts**	**OCR** **(pmol/(s × 10^6^)** **−V**	**OCR** **(pmol/(s × 10^6^)** **+V**	**OCR** **(pmol/(s × 10^6^)** **−V/+H_2_O_2_**	**OCR** **(pmol/(s × 10^6^)** **+V/+H_2_O_2_**
** Basal OCR **	34.7 ± 4.7	34.1 ± 3.5	19.9 ± 2.3	19.0 ± 2.3
** Proton Leak **	12.6 ± 1.4	14.01 ± 0.99	9.1 ± 1.2	11.5 ± 1.4
** Maximal OCR **	59.6 ± 10.3	75.7 ± 8.9 **	23.1 ± 3.3	41.0 ± 5.6 **
**Residual OCR**	5.10 ± 0.69	6.03 ± 0.77	6.77 ± 0.56	5.93 ± 1.08
**Myotubes**	** OCR ** **(pmol/(s × 10^6^)** ** −V **	** OCR ** **(pmol/(s × 10^6^)** ** +V **	** OCR ** **(pmol/(s × 10^6^)** ** −V/+H_2_O_2_ **	** OCR ** **(pmol/(s × 10^6^)** ** +V/+H_2_O_2_ **
** Basal OCR **	38.0 ± 5.5	36.5 ± 2.3	30.9 ± 0.2	25.3 ± 2.5
** Proton Leak **	8.6 ± 1.1	9.6 ± 1.6	9.9 ± 1.1	8.0 ± 1.9
** Maximal OCR **	140.1 ± 13.6	158.6 ± 18.1 *	69.8 ± 6.6	87.3 ± 5.7 *
**Residual OCR**	6.0 ± 2.1	6.3 ± 1.1	5.8 ± 0.6	5.6 ± 1.0

**Table 2 ijms-24-15276-t002:** Flux control ratios derived from high-resolution respiratory measures are reported as average values ± SD from four independent experiments performed in duplicate on C2C12 myoblasts and myotubes. Statistical analysis was performed using Student’s *t*-test (* *p* ≤ 0.05, ** *p* ≤ 0.01; verbascoside-treated (+V) vs. untreated cells (−V)).

**Myoblasts**	**−V**	**+V**	**−V/+H_2_O_2_**	**+V/+H_2_O_2_**
**SRC**	1.72 ± 0.15	2.60 ± 0.22 **	1.60 ± 0.20	2.14 ± 0.08 *
**RCR**	5.33 ± 0.59	5.53 ± 0.78	3.24 ± 0.17	4.11 ± 0.54
**CE**	0.61 ± 0.05	0.57 ± 0.07	0.53 ± 0.06	0.50 ± 0.02
**Myotubes**	**−V**	**+ V**	**−V/+H_2_O_2_**	**+V/+H_2_O_2_**
**SRC**	3.8 ± 0.3	4.3 ± 0.3 *	2.24 ± 0.2	3.6 ± 0.2 *
**RCR**	16.5 ± 1.0	16.9 ± 1.1	7.0 ± 0.3	11.6 ± 1.6 *
**CE**	0.76 ± 0.03	0.78 ± 0.005	0.67 ± 0.04	0.77 ± 0.04

## Data Availability

The data presented in this study are available on request from the corresponding author.

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
