# Peer review of "Verbascoside Elicits Its Beneficial Effects by Enhancing Mitochondrial Spare Respiratory Capacity and the Nrf2/HO-1 Mediated Antioxidant System in a Murine Skeletal Muscle Cell Line"

_ijms, 2023, doi:10.3390/ijms242015276_

Round 1

Reviewer 1 Report (New Reviewer)

Summary:

The authors have described functional evidence in C2C12 murine myoblasts and myotubes that Verbascocide improves Mitochondrial spare respiratory capacity and maximal oxygen consumption rate.   

Authors state that anti-ROS effects and cell viability rescue is likely through the activation of Nrf2/Ho-1/ Pgc1a cellular pathway.   

Major

Overall:  The functional methods to show that Verbascocide improves Maximal OCR, SRC, cellular viability and generates anti-ROS effects are good tools for evaluating benefits of Verbascocide. However, demonstrating this conclusion in the C2C12 cell line alone leaves a lot to be desired.  C2C12 is not a physiologically suitable cell model of mitochondrial deficit, muscular dystrophy/disorder or cachexia. Oxidative stress experiments are only suitable to a certain extent. I would recommend including diseased cell models and/or an in vivo demonstration of findings that translate in vitro to in vivo. 

Secondly,  I believe the mechanistic data on cellular pathway is lacking.  The Nrf2 western blot data is not compelling and must be repeated to make strong conclusions on cellular pathway.  With respect to Figures 5 A and B, authors have demonstrated  plots that indicate significant increases in Nrf2 levels, however, the western blots for total Nrf2 do not indicate this.  Hence, it is not convincing that the authors have quantitated the total Nrf2 protein levels and concluded that the levels have increased under the +V/+H2O2 condition relative to -V/+H2O2.  Authors should repeat experiments and produce more clear western blot data (plots and visual) to support their conclusions on Nrf2 as a target for Verbascocide.   

Minor:

1. A recommendation: For organization purpose and better flow, the order of data should be Figure 1 first as it stands currently, Figure 2 and Figure 4 results should be discussed next as they go together, followed by Tables 1 and 2.  This can be followed by the viability and ROS/ Fluorescence intensity experiments (Figure 4) and finally the western blotting (Figure 5) and IF experiments (Figure 6) for better flow. qPCR experiments for HO-1 and Pgc1a can be Figure 7 but the levels of Pgc1a expression across differentiation does not help the flow of the main data much and should either be a supplemental figure or part of Figure 7.    

Author Response

Reviewer 1

Summary:

The authors have described functional evidence in C2C12 murine myoblasts and myotubes that Verbascocide improves Mitochondrial spare respiratory capacity and maximal oxygen consumption rate.   

Authors state that anti-ROS effects and cell viability rescue is likely through the activation of Nrf2/Ho-1/ Pgc1a cellular pathway.   

Major

Overall:  The functional methods to show that Verbascocide improves Maximal OCR, SRC, cellular viability and generates anti-ROS effects are good tools for evaluating benefits of Verbascocide. However, demonstrating this conclusion in the C2C12 cell line alone leaves a lot to be desired.  C2C12 is not a physiologically suitable cell model of mitochondrial deficit, muscular dystrophy/disorder or cachexia. Oxidative stress experiments are only suitable to a certain extent. I would recommend including diseased cell models and/or an in vivo demonstration of findings that translate in vitro to in vivo. 

We thank the Reviewer for her/his interesting suggestions and definitely we intend to follow this up in our future work. The intent of the present study was to verify whether verbascoside ameliorates the mitochondrial function in healthy cells in normal and oxidative conditions. To this purpose, we chose the C2C12 cell line, as it is a cell model widely used in the literature (see for example Nicholls DG, Darley-Usmar VM, Wu M, Jensen PB, Rogers GW, Ferrick DA. Bioenergetic profile experiment using C2C12 myoblast cells. J Vis Exp. 2010 Dec 6;(46):2511. doi: 10.3791/2511).

Secondly, I believe the mechanistic data on cellular pathway is lacking.  The Nrf2 western blot data is not compelling and must be repeated to make strong conclusions on cellular pathway.  With respect to Figures 5 A and B, authors have demonstrated plots that indicate significant increases in Nrf2 levels, however, the western blots for total Nrf2 do not indicate this.  Hence, it is not convincing that the authors have quantitated the total Nrf2 protein levels and concluded that the levels have increased under the +V/+H2O2 condition relative to -V/+H2O2.  Authors should repeat experiments and produce more clear western blot data (plots and visual) to support their conclusions on Nrf2 as a target for Verbascocide.   

In agreement with the Reviewer’s suggestions, we replaced Figure 5E with another representative blot, which better shows increased protein level of total Nrf2 in C2C12 myoblasts under the +V/+H2O2 condition relative to -V/+H2O2. To facilitate the Reviewer’s inspection, we provided the original images of the blots included in the manuscript with the densitometry profiles of the Nrf2 and tubulin bands, performed with the software UVITEV Nine-Alliance (please, see attached file).

Minor:

A recommendation: For organization purpose and better flow, the order of data should be Figure 1 first as it stands currently, Figure 2 and Figure 4 results should be discussed next as they go together, followed by Tables 1 and 2.  This can be followed by the viability and ROS/ Fluorescence intensity experiments (Figure 4) and finally the western blotting (Figure 5) and IF experiments (Figure 6) for better flow. qPCR experiments for HO-1 and Pgc1a can be Figure 7 but the levels of Pgc1a expression across differentiation does not help the flow of the main data much and should either be a supplemental figure or part of Figure 7.    

We changed the organization of the manuscript, according to the Reviewer's suggestion, and moved Figure 8 of the original manuscript to supplemental material.

Reviewer 2 Report (New Reviewer)

See attached file

Author Response

 ijms-2628548

Verbascoside Elicits its Beneficial Effects by Enhancing Mito-chondrial Spare Respiratory Capacity and the Nrf2/HO-1 Me-diated Antioxidant System in a Murine Skeletal Muscle Cell Line

Summary

The present manuscript brings the hypothesis that Verbascoside can modulates mitochondrial function in skeletal muscle cells and decrease muscle weakness. Despite being an interesting theme, the manuscript presents many discrepancies with literature data, which need to be rethought or more experiments need to be done to reinforce the data presented and refute the previous data from the literature.

Major:

1. The authors use cells culture, to try study the effects of Verbascoside to minimize sarcopenia and many forms of muscular dystrophy associate to mitochondria dysfunction. However, the authors did not shown any result associating the effects of the Verbascoside on sarcopenia, or muscular dystrophy, or muscular fatigue. This study could be better designed.

The intent of the present study was to verify whether verbascoside ameliorates the mitochondrial function in healthy cells in normal and oxidative conditions. To this purpose, we chose the C2C12 cell line, as it is a cell model widely used in the literature (see for example Nicholls DG, Darley-Usmar VM, Wu M, Jensen PB, Rogers GW, Ferrick DA. Bioenergetic profile experiment using C2C12 myoblast cells. J Vis Exp. 2010 Dec 6;(46):2511. doi: 10.3791/2511). In the next future, we will extend our experiments to available animal models of muscle diseases (i.e.  mdx, for Duchenne muscular dystrophy; myd, for congenital muscular dystrophy type 1D; dy mice, for merosin-deficient congenital muscular dystrophy), characterized by muscle weakness and mitochondrial dysfunction.

2. Using cell culture, why the authors didn't permeabilize the cell with saponin or digitonin and why didn't you use a substrate for complex?

The answer to this question can be found within the discussion of our manuscript, which we report here in full: “HRR measurements were performed on intact C2C12 myoblasts and myotubes under physiological substrate supply by using a coupling control protocol [32], which allows to evaluate respiratory function without the addition of exogenous substrates and ADP. This method has proven to be particularly suitable for preserving all mitochondrial perturbations affecting cellular respiration [22,32-34].”

3. In figure 3A and 3B, the authors mention at page 6 line 165 that verbascoside attenuates the cytotoxic effect induced by H2O2, however the authors did not mention against which group and how % was the effect.

We thank the Reviewer for suggesting that. We have now improved the description of cell viability experiments, by adding the following sentences: “indeed, in control myoblasts, exposure to H2O2 reduced cell viability to 69%, whereas 150 mM verbascoside restored it to 88%” and “In particular, exposure of control myotubes to H2O2 lowered cell viability to 16%, while 75 mM and 150 mM verbascoside restored cell viability to 38% and 44%, respectively.”

4. Some subitems titles, are not related with the results descripted. These happens in more than on item.

We have carefully re-checked the paragraph subtitles for any inconsistencies.

5. The order which the authors present the results and figures is very confusing.

As also suggested by Reviewer 1, in order to improve the clarity in the way we present our data, we changed the manuscript organization as follows: Figure 1 first as it stands currently, Figure 2 and Figure 4 (which became Figure 3 in the revised version) results are discussed next as they go together, followed by Tables 1 and 2. This is followed by the viability and ROS/ Fluorescence intensity experiments (Figure 4) and finally the western blotting (Figure 5) and IF experiments (Figure 6). qPCR experiments for HO-1 and Pgc1a are reported in Figure 7, while the levels of Pgc1a expression across differentiation are represented in a separate figure included in supplemental material.    

6. The authors starts the discussion with muscular dystrophy, and make a link with mitochondrial dysfunction, however the authors not show any result that link effectively muscle weakness and mitochondrial dysfunction in C2C12 cell culture

The link between muscle weakness and mitochondrial dysfunction is well documented in the literature and C2C12 cells are commonly used as an in vitro model of skeletal muscle. Treatment with H2O2 mimics the oxidative stress to which muscle tissue is subjected during an intense physical activity or in pathological conditions, such as muscular dystrophy. Therefore, we believe that C2C12 cells represent a reliable, albeit simplified, model for our study. In the future, we will certainly try to extend our studies to more complex models.

Reviewer 3 Report (New Reviewer)

In summary, this research provides valuable insights into how verbascoside can benefit skeletal muscle health by improving mitochondrial function and enhancing antioxidant responses, particularly in myoblasts. These findings have potential implications for the development of therapies for diseases characterised by muscle weakness and mitochondrial dysfunction.

Major
Comments
: Introduction Clarity: The introduction provides an adequate overview of the background and context of the study. However, it could benefit from more explicit and clear statements of the research aims and significance of the study. A concise explanation of the research hypothesis or objectives would help the reader understand the purpose of the study.
Minor Comments: Clarity of the abstract: The abstract should be more concise and focused, providing a brief summary of the objectives, methods, results and implications of the study. Avoid introducing new information or concepts in the summary.

Author Response

In summary, this research provides valuable insights into how verbascoside can benefit skeletal muscle health by improving mitochondrial function and enhancing antioxidant responses, particularly in myoblasts. These findings have potential implications for the development of therapies for diseases characterised by muscle weakness and mitochondrial dysfunction.

Major Comments: Introduction Clarity: The introduction provides an adequate overview of the background and context of the study. However, it could benefit from more explicit and clear statements of the research aims and significance of the study. A concise explanation of the research hypothesis or objectives would help the reader understand the purpose of the study.

We thank the Reviewer for her/his comments and we tried to better explain our hypothesis and the aims of our study within the Introduction. Moreover, we added some sentences to better describe possible therapeutic applications of verbascoside in all pathophysiological conditions characterized by muscle weakness and mitochondrial dysfunction.

Minor Comments: Clarity of the abstract: The abstract should be more concise and focused, providing a brief summary of the objectives, methods, results and implications of the study. Avoid introducing new information or concepts in the summary.

We changed the Abstract according to the Reviewer’s suggestions.

Round 2

Reviewer 1 Report (New Reviewer)

A major concern is that this study provides very limited information on the rescue effect of Verbascocide on mitochondrial function in C2C12 cells under oxidative condition.  For acceptance in IJMS, additional cell lines and/or animal models relevant to models of mitochondrial dysfunction and oxidative stress must be included.  Additionally, the cellular pathway data for p-Nrf and total Nrf is lacking in clarity and not convincing at this time.  

The authors have submitted a new blot for Figure 5E for total Nrf in myoblasts, however it appears that the tubulin band in the +V/+H2O2 condition has only partially transferred onto the blotting membrane. This is not an acceptable image for the conclusion that Total Nrf levels have increased in +V/+H2O2 relative to -V/-H2O2.

Overall, the quality of experimental work conducted in this study has not met the required standard for acceptance into IJMS. While the functional results in C2C12 cells provide an interesting premise, the authors must support this work in physiologically relevant models of disease and mitochondrial dysfunction and inherent oxidative stress.  Also, the quality of western blotting data for total Nrf and Tubulin must be significantly improved.  Hence, the decision to reject this article from acceptance into IJMS has not changed from Round 1 of review.  

Author Response

  1. We replaced Figure 5E with another representative blot (please, see the original figure with the relative densitometry within the attached file).
  2. We agree with the Reviewer that this study paves the way to further and lengthy investigations by using additional cell lines and/or even animal models. However, at the moment, we cannot meet their requests. Based on the interesting results that we have already obtained, we are planning to apply for finding new resources to continue supporting this research line in our laboratory.  

Reviewer 2 Report (New Reviewer)

Dear,

After the revision the manuscript improved, and besides some issues in methodological procedures, the manuscript can be accepted. But I strongly suggest in the future, when working with mitochondrial analysis, if you don't isolate, use some detergent to permeabilize the cells.

Author Response

We thank again the Reviewer for his/her suggestions.

This manuscript is a resubmission of an earlier submission. The following is a list of the peer review reports and author responses from that submission.

Round 1

Reviewer 1 Report

In this manuscript, Sciandra et al. report on the effects of verbascoside on mitochondrial function and expression of specific genes in C2C12 myoblasts in culture. The authors show that verbascoside “enhanced maximal oxygen consumption rate and mitochondrial spare respiratory capacity,” increased Nrf2 protein levels in the presence of H2O2 without affecting Nrf2 transcript levels, increased Nrf2 protein levels near nuclei in both the presence and absence of H2O2, and increased transcript levels of Nrf2 targets HO-1 and PGC-1alpha in both the presence and absence of H2O2. The authors conclude that “activation of the PGC-1alpha/Nrf2/HO-1 signaling pathway might contribute to explain the verbascoside ability to relieve muscular fatigue” with possible implications for muscle function in normal individuals and patients with muscular dystrophy.

Major

Lines 200-1 and figure 5C: The authors state (lines 198-201) that “After exposure of C2C12 to 0.5 mM H2O2 for 2 h as well as after treatment with 150 uM verbascoside for 24 h under non oxidative conditions, Nrf2 was less disperse and predominantly localized near the nucleus, indicating its activation and nuclear translocation (Figure 5C).” This seems to be an overstatement. Panels A and B of figure 5 show that there is more Nrf2 protein present (detected by Western blot) in cells treated with both verbascoside and H2O2 compared with untreated cells and cells treated with either substance on its own. This elevation of total Nrf2 protein must be due to increased protein synthesis and/or decreased protein degradation in the presence of both compounds because Nrf2 mRNA levels were the same for all four conditions. The extent of Nrf2 “activation” was not determined, either by Western determination of the proportion of phosphorylated Nrf2 or by an in vitro assay to measure Nrf2 activity (admittedly, in figure 6, the authors do provide indirect evidence of the latter in vivo with their assays of transcript levels of HO-1 and PGC-1alpha). The amount of Nrf2 “nuclear translocation” was also not measured. Figure 5C (arrows) shows a small number of cells where Nrf2 is “predominantly localized near the nucleus” and not localization within the nucleus. There should be a quantitative analysis of Nrf2 localization within a large number of cells in addition to the example images show in panel 5C. There should also be control data providing reassurance that the higher fluorescence intensity observed near one side of some nuclei is neither due to an optical artifact nor due to uneven staining. The quoted statement above and also the text of the figure legend do not capture what is shown in Figure 5C. According to the sample images shown, there is more Nrf2 near nuclei of cells treated with verbascoside and/or H2O2 (bottom 3 rows) compared with untreated cells (top row). It does not appear that treatment with both verbascoside and H2O2 (2nd row) results in greater accumulation of Nrf2 near nuclei in the sample images shown. Furthermore, to avoid confusion by the reader, the conditions in panel 5C should be shown in the same order (top-to-bottom) as used in other figures (left-to-right): re-order rows verbascoside/H2O2 (top-to-bottom) = ‑/- , +/- , -/+ , +/+ .

Methods Section 4.8 Statistical Analysis (lines 438-442): include what software/version was used for statistical analyses.

Lines 446-7: “…leading to 446 reduced endogenous and H2O2-induced ROS levels…” is speculation.

Minor

The manuscript should be carefully and thoroughly edited. For example, on line 17, “Therfore” should be “Therefore”. In addition to correcting typos, sentence construction should be edited. For example, on lines 37-38 “…muscles obtain the most part of the necessary ATP…” should be something along the lines of “…muscles obtain the most of their ATP…” or perhaps “…muscle cells obtain the most of the ATP necessary for contractile function…”. These are only examples from the first page. The entire manuscript should be reviewed carefully.

Figure 1 and Figure 3. Why not use a continuous x-axis, probably logarithmic?

Author Response

Reviewer 1

Major

  • Lines 200-1 and figure 5C: The authors state (lines 198-201) that “After exposure of C2C12 to 0.5 mM H2O2 for 2 h as well as after treatment with 150 uM verbascoside for 24 h under non oxidative conditions, Nrf2 was less disperse and predominantly localized near the nucleus, indicating its activation and nuclear translocation (Figure 5C).” This seems to be an overstatement. Panels A and B of figure 5 show that there is more Nrf2 protein present (detected by Western blot) in cells treated with both verbascoside and H2O2 compared with untreated cells and cells treated with either substance on its own. This elevation of total Nrf2 protein must be due to increased protein synthesis and/or decreased protein degradation in the presence of both compounds because Nrf2 mRNA levels were the same for all four conditions. The extent of Nrf2 “activation” was not determined, either by Western determination of the proportion of phosphorylated Nrf2 or by an in vitro assay to measure Nrf2 activity (admittedly, in figure 6, the authors do provide indirect evidence of the latter in vivo with their assays of transcript levels of HO-1 and PGC-1alpha). The amount of Nrf2 “nuclear translocation” was also not measured. Figure 5C (arrows) shows a small number of cells where Nrf2 is “predominantly localized near the nucleus” and not localization within the nucleus. There should be a quantitative analysis of Nrf2 localization within a large number of cells in addition to the example images show in panel 5C. There should also be control data providing reassurance that the higher fluorescence intensity observed near one side of some nuclei is neither due to an optical artifact nor due to uneven staining. The quoted statement above and also the text of the figure legend do not capture what is shown in Figure 5C. According to the sample images shown, there is more Nrf2 near nuclei of cells treated with verbascoside and/or H2O2 (bottom 3 rows) compared with untreated cells (top row). It does not appear that treatment with both verbascoside and H2O2 (2ndrow) results in greater accumulation of Nrf2 near nuclei in the sample images shown. Furthermore, to avoid confusion by the reader, the conditions in panel 5C should be shown in the same order (top-to-bottom) as used in other figures (left-to-right): re-order rows verbascoside/H2O2 (top-to-bottom) = ‑/- , +/- , -/+ , +/+ .

Response 1

We thank the Reviewer for their constructive criticism that helped us to improve the manuscript. In particular, we are grateful to the Reviewer for raising the issue about Nrf2 ‘activation’ by phosphorylation. Although, as demonstrated by Bloom and colleagues (Bloom 2003), phosphorylation at serine 40 - and in general at any site of Nrf2 (Sun 2009) - does not affect the ability of Nrf2 to activate transcription of antioxidant genes, it allows Nrf2 to escape inhibition and proteasomal degradation mediated by Keap1, leading to an increased half-life of the protein. To check whether verbascoside exherts its effect by inducing serine 40 phosphorylation, we followed the Reviewer’s suggestion and carried out a set of Western blot experiments to measure the proportion of Nrf2 phosphorylated at serine 40, p-Nrf2 (Ser 40), that we report in the new Figure 5C and 5D of the revised version. This allowed us to observe a higher proportion of p-Nrf2 (Ser 40) under oxidative conditions with and without verbascoside, and to rule out that concomitant treatment with verbascoside and H2O2 results in greater accumulation of p-Nrf2 (Ser 40).

In order to quantify the extent of nuclear translocation of Nrf2 we collected more confocal fluorescence microscopy images. As can be observed from the new Figure 6 of the revised version, the amount of Nrf2 within the nucleus is increased in cells treated with both verbascoside and H2O2 compared with untreated cells and cells treated with either substance on its own. It is noteworthy that the fluorescence intensity was measured on cells at the same focus plane to avoid any possible optical artifact.

We re-ordered rows verbascoside/H2O2 (top-to-bottom) as suggested: ‑/- , +/- , -/+ , +/+ .

 _Bloom, D.A.; Jaiswal, A.K. Phosphorylation of Nrf2 at Ser40 by protein kinase C in response to antioxidants leads to the release of Nrf2 from INrf2, but is not required for Nrf2 stabilization/accumulation in the nucleus and transcriptional activation of antioxidant response element-mediated NAD(P)H:quinone oxidoreductase-1 gene expression. J. Biol. Chem. 2003, 278, 44675-44682

_ Sun Z, Huang Z, Zhang DD. Phosphorylation of Nrf2 at multiple sites by MAP kinases has a limited contribution in modulating the Nrf2-dependent antioxidant response. PLoS One. 2009 4:e6588. doi: 10.1371/journal.pone.0006588

  • Statistical Analysis (lines 438-442): include what software/version was used for statistical analyses. Methods Section 4.8

Response 2

We added the software name and its version in Methods, Section 4.8

  • Lines 446-7: “…leading to 446 reduced endogenous and H2O2-induced ROS levels…” is speculation.

Response 3

We changed the sentence as follows: “our data suggest that verbascoside-induced activation of PGC-1a/Nrf2/HO-1 signalling pathway might be related to the observed reduction of endogenous and H2O2-induced ROS levels and increased myoblasts viability under oxidative stress conditions. ….”

Minor

  • The manuscript should be carefully and thoroughly edited. For example, on line 17, “Therfore” should be “Therefore”. In addition to correcting typos, sentence construction should be edited. For example, on lines 37-38 “…muscles obtain the most part of the necessary ATP…” should be something along the lines of “…muscles obtain the most of their ATP…” or perhaps “…muscle cells obtain the most of the ATP necessary for contractile function…”. These are only examples from the first page. The entire manuscript should be reviewed carefully.

Response 1

We have carefully reviewed and edited the manuscript.

  • Figure 1 and Figure 3. Why not use a continuous x-axis, probably logarithmic?

Response 2

We thank the Reviewer for pointing out the confusing set of lines. We have now removed the shorter one to avoid ambiguity.

Reviewer 2 Report

I have read the paper by Sciandra et al with initial enthusiasm and interest. The authors describe a stimulating effect of verbascoside on mitochondrial respiration in C2C12 myoblasts which was associated with, as the authors claim, increased activation of NRF2 and its target gene HMOX and an increase in PGC-1a mRNA.

I have a number of major concerns about this study that significantly hinder publication of the manuscript in its current form.

First of all, the novelty of this study is very limited. Indeed, the fact that verbascoside displays anti-oxidant activity has been previously described. Also, as the authors state themself in the introduction, the fact that verbascoside has modulatory potential at the level of the mitochondrion has been described previously. Secondly, the rationale of the study is unclear to me. The authors put a lot of emphasis on skeletal muscle mass regulation in the introduction and discussion of the paper but fail to measure any parameter that is associated with the regulation of muscle mass. Also, in this regard, it is unclear to me why the authors have used C2C12 myoblasts as their model of choice as this is not representative for ''adult'' muscle tissue. Why not differentiate the myoblasts to myotubes and use these as a model system? In addition, the data presentation needs improvement as this is confusing at multiple occasions. For example: why not merge Figure 2 and Fig 4? The way they are presented now is very confusing. Also, Fig 5B speaks of Western blot, but no bands are shown, also the confocal images in Fig 5 in my opinion do not show nuclear translocation of NRF2 into the nucleus (also no quantification is present). Finally, the title of the manuscript is misleading as the current title suggests a causal involvement of PGC-1/NRF2/HO signaling in the verbascoside-induced increase in respiration which is not addressed at all in the manuscript (doing so would significantly enhance the manuscript). All things considered I reject the paper for publication in IJMS.    

Author Response

Reviewer 2

I have read the paper by Sciandra et al with initial enthusiasm and interest. The authors describe a stimulating effect of verbascoside on mitochondrial respiration in C2C12 myoblasts which was associated with, as the authors claim, increased activation of NRF2 and its target gene HMOX and an increase in PGC-1a mRNA.

I have a number of major concerns about this study that significantly hinder publication of the manuscript in its current form.

First of all, the novelty of this study is very limited. Indeed, the fact that verbascoside displays anti-oxidant activity has been previously described. Also, as the authors state themself in the introduction, the fact that verbascoside has modulatory potential at the level of the mitochondrion has been described previously. Secondly, the rationale of the study is unclear to me. The authors put a lot of emphasis on skeletal muscle mass regulation in the introduction and discussion of the paper but fail to measure any parameter that is associated with the regulation of muscle mass. Also, in this regard, it is unclear to me why the authors have used C2C12 myoblasts as their model of choice as this is not representative for ''adult'' muscle tissue. Why not differentiate the myoblasts to myotubes and use these as a model system? In addition, the data presentation needs improvement as this is confusing at multiple occasions. For example: why not merge Figure 2 and Fig 4? The way they are presented now is very confusing. Also, Fig 5B speaks of Western blot, but no bands are shown, also the confocal images in Fig 5 in my opinion do not show nuclear translocation of NRF2 into the nucleus (also no quantification is present). Finally, the title of the manuscript is misleading as the current title suggests a causal involvement of PGC-1/NRF2/HO signaling in the verbascoside-induced increase in respiration which is not addressed at all in the manuscript (doing so would significantly enhance the manuscript). All things considered I reject the paper for publication in IJMS.    

  1. At the best of our knowledge, there are not previous studies reporting the beneficial effects of verbascoside on the respiratory function of the mitochondria in myoblasts.  
  2. We agree with the Reviewer that the study of the effects of verbascoside on mitochondrial function in differentiated myotubes would be very interesting. Indeed, this will be the focus of our future investigations. Given the high relevance of the regeneration processes in healthy and diseased skeletal muscle, this first study was aimed at verifying if verbascoside might have beneficial effects on mitochondrial function in C2C12 myoblasts that are considered a good in vitro model of muscle satellite cells.
  3. We did not merge Figure 2 and Figure 4, but in Figure 4 we added the label “H2O2” to clarify that Figure 4 refers exclusively to the samples treated with H2O2.
  4. In Figure 5 we added a panel (Figure 5C) showing a representative Western blot experiment and modified the legend accordingly.
  5. We thank the Reviewer for their insight into this important point. We collected more confocal fluorescence microscopy images in order to properly quantify the extent of nuclear translocation of Nrf2. As can be observed from the new Figure 6 of the revised version, the amount of Nrf2 within the nucleus is increased in cells treated with both verbascoside and H2O2 compared with untreated cells and cells treated with either substance on its own. It is noteworthy that the fluorescence intensity was measured on cells at the same focus plane to avoid any possible optical artifact.
  6. We changed the title of the manuscript as follows: “Verbascoside Enhances Mitochondrial Spare Respiratory Capacity with Concomitant Activation of the PGC-1a/Nrf2/HO-1 Signalling Pathway in Murine Myoblasts”. In order to emphasize that our experiments cannot demonstrate a cause-effect relationship between the activation of the PGC-1a/Nrf2/HO-1 signaling pathway and the antioxidant effect elicited by Nrf-2, we also changed the main text as follows:

Lines 514-517:  “In our study, verbascoside triggered the activation of transcriptional factor Nrf2 and its target genes, HO-1 and PGC-1a, leading to a reduction of endogenous as well as H2O2-induced ROS and to an increased viability of C2C12 myoblasts subjected to oxidative stress” was changed in “In our study, verbascoside triggered nuclear translocation of the transcription factor Nrf2 and upregulation of its target genes, HO-1 and PGC-1a. This suggests that verbascoside might elicit its antioxidant effect by activating the PGC-1a/Nrf2/HO-1 signalling pathway.”

Lines 723-726: “verbascoside activated PGC-1a/Nrf2/HO-1 signalling pathway, leading to reduced endogenous and H2O2-induced ROS levels and to increased myoblasts viability under oxidative stress conditions” was changed in “verbascoside-induced activation of the PGC-1a/Nrf2/HO-1 signalling pathway might be related to the observed reduction of endogenous and H2O2-induced ROS levels and increased myoblasts viability under oxidative stress conditions”.

Reviewer 3 Report

QUESTIONS: 

1-    Any insight into the role of Verbascoside with the peroxisome? Regulation of H2O2? Activation by PPAR? Measures of peroxisomal number/function? 

2-    Why not look at fully differentiated myotubes? Low number of stem cells in adults. 

3-    Any measures of catalase activity? Or MnSOD? Compensatory mechanisms to alleviate oxidative stress? (for figure 2.2 A)

4-    What’s the half-life of Verbascoside? Why only 24-hour incubations?

5-    Any H2O2 preliminary data on why you picked 1mM?

a.     https://www.ncbi.nlm.nih.gov/pmc/articles/PMC2970628/

b.     250uM shown to fragment cells (1hr)

6-    Any measures not in “resting conditions”? Is this physiologically relevant without pushing on the gas or getting the muscles to contract? Energy demand without need?

Author Response

Reviewer 3

QUESTIONS: 

  • Any insight into the role of Verbascoside with the peroxisome? Regulation of H2O2? Activation by PPAR? Measures of peroxisomal number/function? 

We appreciate the Reviewer’s suggestions, but at the moment we are unable to answer these interesting questions.  We will investigate these aspects in our future studies.

  • Why not look at fully differentiated myotubes? Low number of stem cells in adults. 

We agree with the Reviewer that the study of the effects of verbascoside on mitochondrial function in differentiated myotubes would be very interesting. Indeed, this will be the focus of our future investigations. Given the high relevance of the regeneration processes in healthy and diseased skeletal muscle, this first study was aimed at verifying if verbascoside might have beneficial effects on mitochondrial function in C2C12 myoblasts that are considered a good in vitro model of muscle satellite cells.  

  • Any measures of catalase activity? Or MnSOD? Compensatory mechanisms to alleviate oxidative stress? (for figure 2.2 A)

We appreciate the Reviewer’s suggestions, but at the moment we are unable to answer these interesting questions.  We will investigate these aspects in our future studies.

4-    What’s the half-life of Verbascoside? Why only 24-hour incubations?

It has been demonstrated that the concentration of an aqueous solution of 320 micromolar verbascoside at pH 7 is reduced to 62% after 24 h of incubation (D'Imperio, M., Cardinali, A., D'Antuono, I., Linsalata, V., Minervini, F., Redan, B. W., & Ferruzzi, M. G. (2014). Stability–activity of verbascoside, a known antioxidant compound, at different pH conditions. Food research international66, 373-378). We added a sentence at the beginning of paragraph 2.1 with the related reference.

5-    Any H2O2 preliminary data on why you picked 1mM?

  1. https://www.ncbi.nlm.nih.gov/pmc/articles/PMC2970628/
  2. 250uM shown to fragment cells (1hr)

Except for cell viability experiments and 2’,7’-dichlorofluorescin diacetate assays for ROS detection where 1 mM H2O2 was used, all other experiments were carried out with 0.5 mM H2O2, because in these conditions the effect of verbascoside was more pronounced. We believe that our slightly different results compared with those reported in the paper mentioned by the Reviewer can be due to the different experimental conditions, for example using myoblasts vs low density myotube cells.

6-    Any measures not in “resting conditions”? Is this physiologically relevant without pushing on the gas or getting the muscles to contract? Energy demand without need?

We thank the Reviewer for raising these important points. Indeed, we are planning to investigate the effects of verbascoside on cells treated with the catabolic stressor b-guanidinopropionic acid, which chemically induces a context of energy deprivation. Unfortunately, these are time-consuming experiments and we are not able to perform them for insertion within the reviewed version of the current manuscript, due to the deadline set by the Editor. 

Round 2

Reviewer 2 Report

My initial skepticism around this paper has not changed with the revised version. I still believe the novelty is limited and the experimental model (C2C12 myoblasts) is not suitable to answer the research question.